# One-Pot Process: Microwave-Assisted Keratin Extraction and Direct Electrospinning to Obtain Keratin-Based Bioplastic

**DOI:** 10.3390/ijms22179597

**Published:** 2021-09-04

**Authors:** Elena Pulidori, Simone Micalizzi, Emilia Bramanti, Luca Bernazzani, Celia Duce, Carmelo De Maria, Francesca Montemurro, Chiara Pelosi, Aurora De Acutis, Giovanni Vozzi, Maria Rosaria Tinè

**Affiliations:** 1Department of Chemistry and Industrial Chemistry, University of Pisa, Via G. Moruzzi 13, 56124 Pisa, Italy; elena.pulidori@unipi.it (E.P.); luca.bernazzani@unipi.it (L.B.); chiara.pelosi92@gmail.com (C.P.); mariarosaria.tine@unipi.it (M.R.T.); 2Research Center E. Piaggio, University of Pisa, Largo L. Lazzarino 1, 56126 Pisa, Italy; simone.micalizzi@phd.unipi.it (S.M.); Francesca.montemurro@unipi.it (F.M.); a.deacutis@studenti.unipi.it (A.D.A.); giovanni.vozzi@unipi.it (G.V.); 3Department of Information Engineering, University of Pisa, Via G. Caruso 16, 56126 Pisa, Italy; 4Institute of Chemistry of Organometallic Compounds, National Research Council, Via G. Moruzzi 1, 56124 Pisa, Italy; emilia.bramanti@pi.iccom.cnr.it

**Keywords:** keratin, keratin-based bioplastics, microwave-assisted extractions, electrospinning, circular economy, green chemistry

## Abstract

Poultry feathers are among the most abundant and polluting keratin-rich waste biomasses. In this work, we developed a one-pot microwave-assisted process for eco-friendly keratin extraction from poultry feathers followed by a direct electrospinning (ES) of the raw extract, without further purification, to obtain keratin-based bioplastics. This microwave-assisted keratin extraction (MAE) was conducted in acetic acid 70% *v/v*. The effects of extraction time, solvent/feathers ratio, and heating mode (MAE vs. conventional heating) on the extraction yield were investigated. The highest keratin yield (26 ± 1% *w/w* with respect to initial feathers) was obtained after 5 h of MAE. Waste-derived keratin were blended with gelatin to fabricate keratin-based biodegradable and biocompatible bioplastics via ES, using 3-(Glycidyloxypropyl)trimethoxysilane (GPTMS) as a cross-linking agent. A full characterization of their thermal, mechanical, and barrier properties was performed by differential scanning calorimetry, thermogravimetric analysis, uniaxial tensile tests, and water permeability measurements. Their morphology and protein structure were investigated using scanning electron microscopy and attenuated total reflection-infrared spectroscopy. All these characterizations highlighted that the properties of the keratin-based bioplastics can be modulated by changing keratin and GPTMS concentrations. These bioplastics could be applied in areas such as bio-packaging and filtration/purification membranes.

## 1. Introduction

The rapid urbanization and intensification of anthropogenic activities have led to the production of various kinds of waste products, which have accumulated in the ecosystem contributing to environmental pollution. The meat market, slaughterhouses, and the wool industry generate millions of tons of keratin-rich waste biomasses daily [1]. Among keratin-waste biomasses derived from living organisms (such as wool, feathers, hairs, horns, and hoofs), feathers are the most abundant [2]. The high amount of keratin present in keratin-waste biomasses (90% in poultry feathers) [3] has attracted the interest of many researchers in the last few years, intending to promote their valorization and exploitation of it.

Keratin is one of the most abundant structural proteins [4] and, together with collagen, it is the most important biopolymer in animals [5]. Feather keratin has a secondary structure characterized by 9.38% α-helix, 47.19% β-sheet, 32.25% β-turn, and 11.18% random structures [6]. It has a high sulphur content due to a high amount of cysteine residues (7%), which differentiates it from other structural proteins, such as elastin and collagen [7]. Keratin belongs to the family of fibrous structural proteins in which amino acids form intermediate filaments, tightly packed into supercoiled polypeptide chains, interacting by intramolecular and intermolecular disulfide bridges, and ionic, hydrogen, and hydrophobic bonds. These interactions provide keratin with a high tensile strength, mechanical stability, rigidity, and resistance to chemical, enzymatic, and thermal treatments [8]. All these properties, together with its biocompatibility, make keratin and keratin-based materials suitable in many application fields, such as cosmetics [9] biomedicine [10], tissue engineering [11], and bio-packaging [12].

The large-scale use of keratin extracted from feathers strongly depends on the development of cost-effective and time-efficient extraction methods. In the last few years, several attempts have been made to extract keratin from feathers, mainly based on the use of chemicals to break disulphide bonds. Beta-mercaptoethanol, sodium sulfide, and sodium metabisulphite can dissolve feathers with good conversion percentages [13,14,15]. However, these reagents are highly toxic and harmful. Feathers can be hydrolyzed also by denaturing methods, which include alkaline extraction [16], enzymatic and microbial methods [17,18], dissolution in ionic liquid [19], and superheated water [20]. In all of these cases, the product obtained is mainly a low nutritional value powder containing low molecular weight peptides and amino acids, which is generally used for animal feed production [8]. These raw materials are not suitable for technological applications that require high molecular weight keratin. On this basis, the development of a new eco-friendly and industrial scalable process to extract high molecular weight keratin from feathers is desirable from an economical and environmental point of view.

In this work, we propose an innovative, green extraction configuration to recover keratin from poultry feather wastes, followed by the direct electrospinning (ES) of the raw extracted solution to obtain keratin-based bioplastic materials.

For the keratin extraction we used a microwave (MW) coaxial dipole antenna [21], previously successfully applied by us to extract essential oils from orange peel wastes and different herbs [22,23,24]. To evaluate the advantages and disadvantages of our MW assisted extraction process (MAE), mostly in terms of soluble keratin yields and the final electrospun material properties, we compared MAE and conventional heating (CH) extraction. Besides, the yield of the MAE was optimized by investigating the effects of the extraction time and the ratio solvent/feathers. The composition of the extracted solutions and morphology of the purified keratin were also investigated. The raw extracted keratin solutions (namely REK) were then used to prepare keratin-based materials using an ES process, a technology directly scalable up to an industrial level [25]. The ES process, namely a high-voltage driven spinning process, has been used to produce nonwoven nanofiber mats for multiple applications including, but not limited to, biomedical, electronics, and filtration [26]. Several bio-based polymers can be processed, including cellulose, chitosan, polylactides (e.g., PLA), and polyhydroxyalkanoates for obtaining membranes with a high surface-to-volume ratio and tunable porosity [27,28].

The direct ES of pure keratin solutions is, in general, not straightforward due to their unsuitable rheological and conductive properties. Keratin processability is restricted to blends with appropriate polymers suitable for electrospinning [29]. Specific polymers, such as poly(ethylene oxide), poly(vinyl alcohol), and poly(hydroxybutylate-cohydroxyvalerate), were previously blended with keratin to improve its electrospinning processability [30,31,32,33,34]. In the context of this work, gelatin was selected due to its biodegradability, biocompatibility, and high miscibility with keratin [10,35], and because it is suitable for the ES process.

Gelatin is a multifunctional food ingredient, with its prevalent use being as a gelling agent. In this respect, gelatin from a mammalian source shows the best physicochemical properties, such as gel strength, melting, gelling temperature, and rheological properties. Many studies have been aimed at characterizing the rheological and gelling properties of gelatin solutions, the effect of additives used as co-gelling agents (*i*- and *k*- carrageenan, sodium alginate, acyl-gellan [36,37,38]), and to investigating other gelling agents alternative to gelatin [39,40]. Most of these studies refer to the properties of gelatin (or its substitutes) in solution or in the gel state, investigating structural modifications at a nanoscale with advanced imaging techniques (atomic force microscopy, confocal laser scanning microscopy, small-angle x-ray scattering, and field emission scanning electron microscopy (SEM)), in combination with classical spectroscopic techniques (FTIR), rheological, and zeta potential studies. In the case of electrospun structures, both gelatin-based and keratin-based, most of the studies are instead aimed at characterizing their mechanical properties, gas and water permeability, and morphology, the latter mainly through SEM studies. Interesting information about the secondary structure of electrospun structures is also derived from FTIR studies. In the case of electrospun structures obtained from keratin/PLA, and keratin/poly (ethylene oxide) blends, the presence of β-sheet tends to decrease to the advantage of the α-helix structure compared with films obtained from keratin alone [28,29].

Here, a porcine gelatin solution was blended directly with the REK solution for the ES process, adding 3-(Glycidyloxypropyl)trimethoxysilane (GPTMS) as crosslink [41].

Ultimately, the aim of the work was the investigation of how the GPTMS content, the keratin features, and the amount, affect the thermal, mechanical, and barrier properties of the resulting electrospun materials, without a microscopic evaluation of keratin-gelatin interaction.

## 2. Results

### 2.1. Keratin Extraction and Characterization

Firstly, we investigated the effects of the extraction time, solvent/feathers ratio, and heating mode (MAE or CH) on the extraction yields of keratin. Table 1 shows the percentages for the initial weight of the feathers, converted feathers, non-soluble keratin (NSK), and total-soluble keratin (TSK), obtained under different experimental conditions.

The results show that the yield of converted feathers depended significantly on the extraction time, but only when the conventional heating procedure was used (*p* < 0.05). On the contrary, the time influenced the TSK yield%, which increased by increasing the extraction time in the MAE and CH heating modes (*p* < 0.05). Moreover, we observed that the results were practically unaffected by the solvent/feather ratios (75:1 or 150:1), both in MAE and CH extraction processes. For this reason, we used the 75:1 ratio in the rest of experiments to obtain more concentrated keratin extracts. The analytical protocol described in the experimental section, and summarized in Scheme 1, allowed us to evaluate the composition of the soluble keratin. The results are summarized in Table 2, where the concentration (mg/mL) of TSK in the explored conditions is reported together with its composition in terms of amino acid content and the soluble keratin fractions (SK) for different molecular weights.

Through analyzing the different heating modes, we observed that when comparing 3 h to 5 h of extraction, the SK_<1200_ concentration increased significantly in both processes (*p* < 0.05), but the increase was greater in the MAE process. Conversely, the amino acids concentration increase was greater in the CH process. The HPLC-DAD analysis showed the presence of proline, valine, and tyrosine (Appendix A), likely due to the partial hydrolysis of keratin in 70% *v/v* acetic acid during the extraction process. These values are lower than those reported in the literature for feather hydrolysis in conventional acidic conditions (6 M hydrochloric acid, 100 °C) [42,43] according to the milder extraction conditions used in our process. The molecular weight distribution of the soluble keratin was analyzed by SDS-PAGE (Appendix A). More in detail, the TSK fraction, dried under N_2_, was resolubilized in an aqueous buffer. Part of the keratin residue was insoluble, and the soluble keratin portion showed a molecular weight distribution between 5000–10,000 Da. In addition, we studied the morphology of the soluble keratin fraction with the highest molecular weight (SK_>7000_) using SEM microscopy. Figure 1 shows a representative SEM image of dried SK_>7000_ obtained after 2 h of MAE. We observed that keratin is arranged in compact laminar structures. The effect of these features on the following applications will be extensively addressed in Section 3.

### 2.2. Mechanical, Barrier and Morphology Properties of Keratin-Based Materials

All the blends processed using ES have produced successful results. Figure 2a represents the material obtained using the G_10_K_0.08_(2hMAE)GPTMS_3_ blend. The results obtained from the uniaxial tensile tests, water permeability, and SEM analyses, are reported in Table 3.

The uniaxial tensile tests indicated that the keratin-based bioplastics were significantly stiffer with respect to gelatin-only bioplastic, exhibiting a lower failure strain, higher elastic modulus, and lower toughness (*p* < 0.05). When comparing the bioplastic materials with different amounts of GPTMS, a higher concentration of the crosslink made them stiffer (higher elastic modulus) but more brittle (lower toughness). For example, by increasing the amount of GPTMS from 3% to 6%, the bioplastics prepared with keratin extracted with a 2 h MAE process showed +40% increase in elastic modulus (from about 160 to 230 MPa), but a 67% decrease in toughness (from about 46 to 16 kJ·m^−3^).

With regards to the effect of the heating mode, keratin extracted through the MAE process lead to tougher biomaterials than those obtained through the CH mode (*p* < 0.05); this result is more evident for the materials with 3% GPTMS.

The results of the water permeability tests show that the presence of keratin lessened the water permeability of the bioplastics. For example, when adding the keratin extracted with a 2 h MAE process to gelatin, the Darcy permeability K decreased 10 times from 22 to 2 × 10^−15^ m^2^. Moreover, it can be observed that the keratin extracted after 2 h lead to a greater decrease in water permeability than the keratin extracted after 5 h. However, this effect was observed only in the MAE process.

Figure 2b and c show the SEM images of G_10_K_0.08_(2hMAE)GPTMS_3_ and G_10_K_0.08_(2hMAE)GPTMS_6_ keratin-based bioplastics. The different biomaterials exhibited common features consisting of a fibrous structure (see also Appendix A). The fibers are smooth, bead-free, and randomly oriented for all the gelatin/keratin samples, with their morphology and diameter being influenced only by GPTMS cross-linker concentration. The highest cross-linker concentration (6% *v/v*) lead to fused fibers with a belt-like morphology and a larger diameter (Table 3), while the lowest (3% *v/v*) lead to straight fibers with circular cross-section.

### 2.3. Physical-Chemical Properties of Keratin-Based Materials

The physical-chemical properties of the bioplastic materials were evaluated by TGA, DSC, and FTIR spectroscopy. The results are summarized in Table 4.

The thermogravimetric analysis shows that all the keratin-based bioplastic materials had a similar degradation pattern during the heating scan (Appendix A). Figure 3a shows representative thermogravimetric curves of G_10_K_0.08_(2hMAE)GPTMS_3_ and G_10_K_0.08_(2hMAE)GPTMS_6_ materials. Mass losses occurred in two different regions: in the first one, below 100 °C, the evaporation of adsorbed water took place, while in the second region, between 150 °C and 600 °C, the protein structure decomposed. The DTG curves evidenced that the protein decomposition occurred in two overlapped steps with maximum degradation rates at approximately 330 °C and 430 °C.

Table 4 illustrates the onset temperature, weight losses, and residue at 900 °C for all the keratin-based bioplastic materials considered. The keratin and the cross-linker concentration seemed to have little influence on thermal stability. The temperatures at which degradation began (T_onset_) in bioplastics containing keratin were, indeed, lower than in bioplastics composed by gelatin only, while those containing a higher amount of GPTMS showed a higher T onset.

The DSC analysis highlighted the effects of the different extraction times and cross-linker concentrations on the thermal properties of bioplastic materials, particularly on the glass transition temperature (T_g_). The DSC curves were analogous for all bioplastic materials (Appendix A).

The bioplastic materials composed of a higher amount of GPTMS presented a higher T_g_ (Table 5). The effect of the type of keratin is peculiar, as the use of keratin obtained after 2h of extraction caused a decrease in T_g_, while the use of keratin obtained after 5 h of extraction caused an increase in T_g_. For example, when comparing G_10_GPTMS_3_, G_10_K_0.08_(2hMAE)GPTMS_3_ and G_10_K_0.16_(5hMAE)GPTMS_3_ (Figure 3b) the T_g_ changed from 89.6 °C to 74.3 °C to 95.4 °C, respectively.

Appendix A shows the ATR-FTIR spectra of the gelatin powder, SK_IP_ (obtained from 2h MAE and 5h MAE extracts), and all of the bioplastic materials in the 4000–600 cm^−1^ regions. The ATR-FTIR spectrum of GPTMS (Appendix A) is reported as a comparison and presents the typical absorptions of Si-O and C-O stretching vibrations. The spectra of gelatin, SK_IP,_ and bioplastics have the characteristic absorptions of proteins, namely amide I in the 1700–1600 cm^−1^ region (due to the C=O stretching vibration), amide II in the 1600–1480 cm^−1^ region (due to the coupling of the N-H in plane bending and C-N stretching modes), and amide III in the 1350–1190 cm^−1^ region (due to C-N stretching coupled to the in-plane N-H bending mode).

Table 4 summarizes the ratio of optical densities of amide I and amide II bands (AII/AI) and of two bands characteristic of the CH_2_ bending (δ1447/δ1400) of bioplastic samples.

All bioplastics materials containing 6% GPTMS showed the lowest values of the AII/AI ratio (0.619 ± 0.003 average value of G_10_K_0.08_(2hMAE)GPTMS_6_, G_10_K_0.16_(5hMAE)GPTMS_6_, G_10_K_0.08_(2hCH)GPTMS_6_, G_10_K_0.16_(5hCH)GPTMS_6_), which was significantly different from the AII/AI ratio of the corresponding bioplastic materials containing 3% GPTMS (0.637 ± 0.010).

Higher values of AII/AI ratio were observed for gelatin powder (0.714 ± 0.014) and SK_IP_ (e.g., 0.808 ± 0.016 and 0.798 ± 0.016 for SK_IP_ 5h MAE and SK_IP_ 2 h MAE, respectively).

Moreover, all bioplastic samples exhibited higher values of the δ1447/δ1400 ratio.

### 2.4. Statistical Analysis

For a better understanding of the influence of each parameter on the global properties of keratin-based materials, we carried out a principal component analysis (PCA), including the parameters obtained from the uniaxial tensile tests, water permeability tests, SEM, TGA (T_onset_), DSC (T_g_), IR (AII/AI and δ1447/δ1400 ratios), and the concentrations of all fractions identified in the REK extracts (amino acids, SK_<1200_, SK_1200–1700_, SK_>7000_). The PCA allowed us to reduce the number of variables which describe the system, highlighting the correlation among different material properties and the starting blend composition. Figure 4 reports the biplot obtained from this statistical analysis.

The biplot highlights a separation of the observations on the first component (F1) as a function of the keratin content. Obviously, the materials that do not contain keratin are arranged at negative value of F1 (G_10_GPTMS_3_ and G_10_GPTMS_6_).

The separation of the observations on the second component (F2) is a function of the GPTMS concentration. The materials containing lower GPTMS amount (3%) are arranged at negative values of F2, while those containing 6% GPTMS are placed at positive values of F2. The influence of keratin and GPTMS content on the final material properties, highlighted by PCA, will be discussed in detail in Section 3.

## 3. Discussion

We propose a new procedure to extract keratin from waste feathers in mild conditions, using acetic acid as a reaction medium.

The results highlight the influence of different parameters (the solvent/feather ratio, extraction time, and MAE or CH heating mode) on the keratin yield. We found that higher extraction times were more efficient, as the total amount of soluble keratin increased going from 2 to 5 h (Table 2). The different heating modes seemed to influence only the composition of the extracted solution: MAE decreased the amino acid content while increasing the content of soluble proteins and, particularly, the low molecular weight SK_<1200_ fraction.

Moreover, the analysis using SEM on the dried SK_>7000_ fraction (Figure 1) provided us insight into the protein arrangement following the extraction process, and showed that the protein structures were organized in peculiar multilayer aggregates. This keratin rearrangement could explain why its solubilization is often difficult. When the TSK, dried under N_2_, was resolubilized in aqueous buffer to perform SDS PAGE, part of the keratin residue became insoluble, likely due to the formation of insoluble, hydrophobic keratin aggregates.

Table 5 shows a comparison of our results with the most efficient procedures reported in the literature in different experimental conditions, including ours.

**Table 5 ijms-22-09597-t005:** Summary of the literature data related to extraction procedures of keratin from waste feathers.

Extraction Method	Extraction Conditions	Yields
Alkaline/MAE [44]	NaOH 0.5M 800W Solid: liquid (*w/w*) = 1:50 Temperature (°C) = n.r Time = 10 min	Thiol = 24.72 mM Protein (Folin Ciocalteau) = 26.74 mg/mL Amino Acid (Ninhydrin Assay) = 69.4 mg/g Molecular Weight = n.r
Reductive [14]	Na_2_S (500mM) Solid: liquid (*w/w*) = 1:40 Temperature (°C) = 50 Time = 6 h	Keratin yield % = Extracted keratin/feathers (*w/w*) = 80% Protein (Bradford assay) = 1.6 mg/mL Molecular Weight = n.r
Sulphitolysis [28]	Urea 8M; Na_2_S_2_O_5_ 0.2M; SDS/feathers 0.6; NaOH up to pH 6.5 Solid: liquid (*w/w*) = 1:36 Temperature (°C) = 65 Time = 5 h	Keratin yield % = (Feathers-unconverted feathers)/(feathers) = 87.6% Molecular Weight (SDS-PAGE) = 11–20, 32, 37, 50, 75 kDa
Enzymatic and microbial [17]	Keratinases-Bacillus subtilitis pH 8 Temperature (°C) = 28 Time = 5 days	Protein (Lowry method) = 95% Molecular Weight (MALDI-TOF) = 0.8–1.1 kDa
Ionic liquid [19]	[Bmim][Cl]^+^10%Na_2_SO_3_ Solid: liquid (*w/w*) = 1:20 Temperature (°C) = 90 Time = 1 h	Keratin yield % = Extracted keratin/feathers (*w/w*) = 75.1% Protein (hydrolysis HCl-sum amino acids) = 72% Molecular Weight (GPC) = 8.83–9.74 kDa
Acid [45]	H_2_O-H_2_SO_4_ 1:1 Temperature: 50 °C pH: 4–5 Time: 12 h Recrystallization in EtOH/H_2_O) with microwaves or ultrasounds	Not reported
Acid/MAE (This work)	Acetic acid 70% *v/v* 90 W Solid: liquid (*w/w*) = 1:75 Temperature (°C) = 104.6 Time = 5 h	Converted feathers % = (Feathers-unconverted feathers)/Feathers = 38 ± 5% NSK = 12 ± 2% - TSK = 26 ± 1% Amino acids = 0.94 ± 0.02 mg/mL, SK_<1200_ = 0.80 ± 0.03 mg/mL SK_1200–7000_ = 0.87 ± 0.02 mg/mL, SK_>7000_ = 0.49 ± 0.04 mg/mL Molecular Weight (SDS-PAGE) = 5–10 kDa

Examining the methods shown in Table 5 from the point of view of the actual implementation of a green one-pot process of keratin extraction, and direct electrospinning, we observed that the process developed by Lee et al. [44] was very fast (10 min), but no information on the molecular weight of the keratin extracted was reported (parameter that influence the ES process). Sharma et al. [14] reported a keratin extraction method with a high yield of keratin extraction without specifying its molecular weight. The enzymatic method proposed by Villa et al. [17] allows one to obtain a high yield of keratin but with a low molecular weight. The methods developed by Isarankura Na Ayutthaya et al. [28] and Ji et al. [19] allow one to obtain high keratin extraction yields with a high molecular weight (useful for ES process), but the many chemicals used could compromise the following ES. Conversely, Perez-Gutierrez and their coworkers [45] proposed the extraction of keratin within mild acidic conditions (with a subsequent recrystallization in ethanol/water with microwaves and ultrasounds), evaluating the morphology of the final material, without reporting the extraction yield or the molecular weight of the extracted keratin. Even when considering the diversity of the experimental conditions, and of the techniques used to analyze the extracted keratin, we observed that, in our conditions, the keratin yields were lower than those reported by the other authors. However, the method proposed in this work has several advantages, the most important being the possibility to directly process the REK solutions to obtain, without any purification step, the keratin-based bioplastic through ES. The possibility of using a one-pot process is very relevant considering keratin’s tendency to form insoluble aggregates during the purification steps (as shown by the SEM and SDS-PAGE analysis), making subsequent manipulations more difficult.

Therefore, the one-pot process developed in our work is innovative from an economic, industrial, and environmental point of view. Furthermore, compared to the extraction approaches described in previous studies, it represents a greener solution that should improve the feasibility of any possible application.

To explore how the keratin content and its molecular weight affect the mechanical and physical-chemical properties of the final keratin-based materials, we directly used all REK solutions obtained in different extraction conditions (extraction time and heating mode) to prepare different electrospun materials, evaluating the effect of these parameters on the final properties of these biomaterials.

Although our goal was to produce keratin-based bioplastics through ES, keratin-only solutions are known to be not commonly electrospinnable due to their conductivity and unsuitable rheological properties. For these reasons, suitable blends of waste-derived keratin and porcine gelatin were used in our study, together with the appropriate crosslinking agent (GPTMS) [34]. The properties of gelatin-only electrospun materials were taken as a reference to study the effects of added keratin. All of the materials have a fibrous structure (SEM results), which illustrated that no phase separation between keratin and gelatin occurred during the ES process, thereby supporting the compatibility of the two biopolymers [35]. Furthermore, a higher percentage of GPTMS led to the fusion of different fibers together, producing fibers characterized by belt-like morphologies which were also observed in the electrospun materials of wool keratin and fibroin [46], and those of feather keratin and PVA [47]. Koombhongse et al. [48] have explained this phenomenon as the irregular evaporation of the solvent from the core of the jet during the electrospinning process, which caused the filament to collapse.

All of the samples presented the same thermal degradation pattern in which the protein structure collapsed in two steps, the first at about 330 °C and the second at 430 °C. The presence of the second mass loss, together with IR data, confirmed the successful polymerization, indicating the presence of compounds with a higher molecular weight which degrade at higher temperature. As the GPTMS content increased, this second mass loss and the residue at 900 °C also increased along with the T_g_ value determined by DSC, which moved toward a higher temperature accounting for a higher crosslinking degree of the biomaterials. This was also evident in the SEM images since different fibers were fused, which produced a more crosslinked network.

The stiffness, toughness, and permeability of the materials can be modulated by changing both the keratin and GPTMS content. The biplot (Figure 4) highlights that amino acids, SK_<1200_, SK_1200–1700_, and SK_>7000_ variables are about inversely correlated with K Darcy, T_onset_, ε_max_ (especially low molecular weight protein hydrolysis products, i.e., SK_<1200_ and amino acid content), and U (especially higher molecular weight keratin, i.e., SK_1200–1700_ and SK_>7000_ fractions). Therefore, keratin decreases the water permeability, the thermal stability, the failure strain, and the toughness. Conversely the elastic modulus and the keratin fractions (amino acids, SK_<1200_, SK_1200–1700_, SK_>7000_) are directly correlated and, accordingly, keratin increases the bioplastic’s stiffness. The ability of keratin to make the materials stiffer is observed also by Ming et al. [47] in the keratin/PVA electrospun materials, and a possible explanation could be related to the β-sheet structure of keratin.

The GPTMS seemed to affect mainly the thermal stability, glass transition temperature, fibers diameter, toughness, and AII/AI and δ1447/δ1400 ratio. In particular, the materials that contained more GPTMS were characterized by higher thermal stability, glass transition temperature, and fiber diameter, while those with a lower GPTMS concentration presented a higher toughness, AII/AI, and δ1447/δ1400 ratio. Materials containing keratin were more stiff and less tough than those containing only gelatin, and even an increase in the concentration of GPTMS (from 3% to 6% *v/v*), with the same keratin content, produced the same effect (see uniaxial tensile test results).

The presence of keratin decreased the water permeability (lower K Darcy) of the bioplastics. Electrospun materials with differing permeability can be obtained by changing the composition and the heating mode used to obtain the REK solution. The MAE-REK solution obtained after 2 h lead to a greater decrease in water permeability than the MAE-REK solution after 5 h (Table 3). We can hypothesize that a greater hydrolysis degree, evidenced by the higher concentration of SK_<1200_ in the 5 h MAE-REK solution (0.80 ± 0.03 vs. 0.45 ± 0.02, Table 2), resulted in an increase of the hydrophilicity responsible for the higher permeability compared to the 2 h MAE-REK solution.

The increase of the SK_<1200_ fraction (0.40 ± 0.05 vs. 0.48 ± 0.04, Table 2) in the CH process was not significant, nor was the k Darcy value.

The effect of the REK solution type on T_g_, evaluated by DSC, was of particular interest. The T_g_ decreased using the REK solution obtained after 2 h of extraction, and increased using the REK solution obtained after 5 h of extraction (especially for MAE-REK), with respect to the material composed only of gelatin. By considering the previous REK solution characterizations in terms of the SK_<1200_, SK_1200–7000_, SK_>7000_, and amino acid composition, a possible explanation for this behavior could be related to the different molecular weights of keratin molecules present in the REK solution and of commercial gelatin. Gelatin has a molecular weight between 50 to 100 kDa, while keratin molecules are about 10 times lower in weight. As a consequence, the branches and ramifications formed during the cross-linking reaction could be shorter than those formed with only gelatin, leading to a higher mobility and, therefore, to a lower T_g_. Instead, using the REK solution obtained after a 5 h extraction (more concentrated than the REK solution obtained after 2 h, and richer in the SK_1200–7000_ fraction, especially for MAE-REK), the crosslinked chains could form a more branched network than that formed with only gelatin, leading to a higher rigidity of the structure which results in an increase of T_g_.

Infrared spectroscopy was one of the earliest experimental methods used to evaluate the secondary structure of polypeptides and proteins [49,50,51]. The amide I peak at 1633 cm^−1^ is generally associated with a β-structure. The frequency of amide II, as well as the ratio of Amide II/Amide I optical densities, were not associated with specific secondary structures, but their variation was suggestive of deep changes in the protein conformation. Moreover, the AII/AI ratio generally increased during denaturation and decreased when the proteins gained an ordered structure [52].

Moreover, the absorptions in the CH_2_ bending region (1500–1300 cm^−1^) were also affected by changes in protein structure. Despite the fact that CH_2_ groups are not commonly investigated for conformational studies, their absorptions are affected during β-sheets formation process, which is in agreement with Maiti et al. [52,53]. The results of the conformational analysis of keratin-based materials reported above (Table 4) are suggestive of highly ordered structures in all bioplastic, likely based on β-sheets conformation induced by the ES process and/or the presence of keratin in the blend. Higher values of the AII/AI ratio suggestive of less ordered molecular structures are, indeed, observed for gelatin powder and SK_IP_.

A deeper inspection of the ATR-FTIR spectra for the study of the protein conformation of starting materials, such as porcine gelatin and extracted keratin, as well as electrospun samples, is in progress and is beyond the aim of this study.

An overview of the procedure and the data discussed above is summarized in “Scheme 2”.

## 4. Materials and Methods

Poultry feathers were provided by Consortium SGS (Santa Croce sull’Arno, Italy), a company that processes animal byproducts. Acetic acid (ACS grade), methanol (HPLC grade), sulphuric acid (ACS grade), gelatin from porcine skin type A (G2500), 3-Glycidyloxypropyl)trimethoxysilane (GPTMS), and benzoylated dialysis tubing MWCO 1200 Da (Code D 2272) were purchased from Sigma-Aldrich (Milan, Italy). Visking dialysis tubing MWCO 7000 Da (Code DTV07000.02) was purchased from Medicell Membranes Ltd. (London, England). The 4x Laemmli sample buffer, 2-mercaptoethanol, 4–20% Mini-PROTEAN^®^ TGX™ Precast Protein Gels, Precision Plus Protein Dual Xtra Standard (2–250 kDa), 1x TGS Buffer (running buffer), Coomassie Brillant Blue R-250 destaining solution, and Bio-safeTM Coomassie G-250 Stain were purchased from Bio-Rad (Milan, Italy).

The keratin-based MAE solution was carried out using a coaxial dipole antenna to apply MW energy inside the extraction medium. The poultry feathers were previously cleaned and defatted through a wet thermal treatment at 80 °C and dried before use. Fibers of 0.2–0.5 cm length were obtained by grinding poultry feathers using an IKA-MF 10 from IKA^®^ (Janke & Kunkel-Str. 10, Staufen, Germany/Deutschland) basic microfine grinder drive (speed range 3000–6500 rpm). The ground feathers and 70% *v/v* acetic acid were loaded in the MW assisted vessel. The vessel was wrapped by a metallic grid to prevent the loss of MW irradiation out of the reaction medium and to ensure safe operating conditions. A magnetic stirrer was placed on the base of the MW-assisted extractor device. MW energy was applied by means of a coaxial dipole antenna immersed into the extraction medium protected by a glass tube. The MW source was a magnetron oscillator that supplies up to 1200 W of continuous MW irradiation power at a frequency of 2450 MHz. A fiber optic temperature sensor (FOT-L NS-967, FISO technologies) was used to control in situ the temperature of the extraction medium. The substrate dispersion was heated using 90 W of MW power and stirred continuously at a reflux temperature (104.6 °C). After the acidic treatment, the broth was sieved with steel sieve (cut off at 500 µm) to separate the unconverted feathers from the raw extracted keratin (REK) solution. The unconverted feathers were washed with water and dried at 60 °C for 24 h to calculate the difference of the converted feathers yield (% *w/w* with respect to initial feathers). The REK solution was used without any further purification for the ES process.

In addition, the solution was fully realized according to the following steps. Firstly, it was centrifuged at 8000 rpm for 15 min. The NSK residue was washed with water until its neutralization and dried at 60 °C to calculate the NSK yield (% *w/w* with respect to the initial feathers). The supernatant was divided into five aliquots. The first one was dried under nitrogen flow and weighed to calculate the TSK yield (mg/mL), and its molecular weights distribution was determined by SDS-PAGE.

The second aliquot was dialyzed using dialysis tubing with 7000 Da MWCO against distilled water for 48 h (distilled water was changed 4 times) and dried to calculate the yield of nominally soluble keratin macromolecules with a molecular weight > 7000 Da (SK_>7000_) (mg/mL). The dialyzed residue was kept in a desiccator and used for SEM analysis. The third aliquot was dialyzed using dialysis tubing with 1200 Da MWCO, according to the procedure described for the second aliquot. The dialyzed product was dried to calculate the yield of soluble keratin macromolecules with a molecular weight > 1200 Da (nominally SK_>1200_) (mg/mL).

The fourth aliquot (nominally SK_<3000_) was ultrafiltered with a Microcon centrifugal filter unit YM-3 membrane (MWCO 3000 Da, 1,4000 rpm, 2 h) and analyzed using a HPLC/DAD for the determination of possible amino acids.

The fifth aliquot (nominally SK_IP_) was employed to purify keratin using isoelectric precipitation (pI = 4.5–5.5) [54] and adding an ammonia solution. After 24 h, the keratin precipitate was separated by centrifugation (14,000 rpm, 15 min), washed with water, and centrifuged (at 1,4000 rpm for 15 min) for two times and finally dried at room temperature to investigate the protein structure through ATR-FTIR spectroscopy. Scheme 1 shows a diagram of the procedure described above employed for the full characterization of the extraction products.

Conventional heating (CH) acidic extraction was carried out under the same experimental conditions as the MAE process, using an oil bath at the reflux temperature (104.6 °C) and while stirring continuously. The extract was then subjected to the same analysis and purification protocol described above.

The effect of the extraction time (2 or 5 h) and solvent/feathers ratio (75:1 or 150:1 mL/g) on the keratin yield was investigated for both in MAE and CH extractions.

An Agilent 1260 Infinity HPLC system (G1311B quaternary pump) equipped with a 1260 Infinity High Performance Degasser, a TCC G1316A thermostat, 1260ALS autosampler (G1329B), and a UV/vis diode array (1260 DAD G4212B) were used. The identification of amino acids on the SK_<3000_ fraction was based on the comparison of the retention time and UV spectra of standard compounds. The 220 nm detection was selected to control the interference of high absorbing compounds. Samples were diluted 10 times in the eluent phase before the injection. The chromatographic separation was carried out using a Zorbax Phenyl-Hexyl RP C18 (Agilent Technology) 250 × 4.6 mm (silica particle size 4 μm) at 45 °C using the following elution profile: 15 min isocratic elution with 100% 5 mM sulphuric acid (pH 2.2), followed by 10 min gradient to 80% methanol and 10 min isocratic elution in 80% methanol (flow 0.8 mL∙min^−1^). The column was rinsed with 100% methanol for 15 min and the re-equilibration step was performed. All of the solutions were filtered using a 0.22 μm regenerate cellulose filter (Millipore, Milan, Italy). Appendix A reports the detected and quantified amino acids.

Sodium dodecyl sulphate-polyacrylamide gel electrophoresis (SDS-PAGE) was performed to analyze the molecular weight distribution of TSK, and 1 mg of each sample was mixed with 350 µL of 4× Laemmli sample buffer, 50 µL of 2.5% *v/v* of 2-mercaptoethanol as reducing agent, and 1000 µL of deionized water. Afterwards, the sample solution was heated at 90 °C for 5 min and 10 µL were loaded into the PAGE (4–20% mini-PROTEAN^®^ TGX protein gels). The precision plus protein Dual Xtra mixture (2–250 kDa) (10 µL) was used as the molecular weights standard. The electrophoretic cell was filled with 1× TGS running Buffer diluted 10 times and then the electrophoresis was carried out at constant voltage of 200 V for 30 min. Finally, the gel was treated with Coomassie Brillant Blue R-250 destaining solution for 60 min to fix the 2 and 5 kDa bands, rinsed in deionized water, and then stained in bio-safe Coomassie G-250 for 90 min. The SDS-PAGE obtained is reported in Appendix A.

The gelatin solution (20% *w/v*) was prepared by mixing gelatin powder in 90% *v/v* acetic acid and stirred at room temperature overnight.

Gelatin/keratin blends were prepared by mixing the REK (70% *v/v* acetic acid) with 20% *w/v* gelatin solution (90% *v/v* acetic acid) in a 1:1 volume ratio, stirring for 15 min at room temperature and subsequently GPTMS cross-linking agent was added. After 40 min of stirring, the gelatin/keratin blends were processed through ES.

To observe the keratin effect on the mechanical properties and permeability of ES materials, 10% *w/v* gelatin solution in 90% acetic acid, with GPTMS and without REK, was also prepared. At first, to reproduce an actual control sample, the 10% *w/v* gelatin solution was prepared in 80% *v/v* acetic acid. Unfortunately, the resulting ES material was not compact and suitable for further studies; therefore, we decide to use 90% *v/v* acetic acid.

To examine the effect of the different extraction conditions (different extraction time and heating mode) on the ES materials performances, we tested the REK solutions obtained both after 2 and 5 h of extraction and for both the MAE and CH processes.

Moreover, two different concentrations of GPTMS were used. Based on the literature data, we tested 6% *v/v* [34] and subsequently decided to also explore a lower concentration, namely 3% *v/v*. The samples’ composition processed through ES are summarized in Table 6.

The ES process was performed using a Linari Engineering apparatus (Linari Eng, Italy) featuring an aluminum foil as a grounded collector, and a commercial 10 mL syringe with 21 G needle as its spinneret. The spinning parameters were as follows: the applied voltage was 50 kV, the distance between the spinneret and the grounded collector was 15 cm. The flow was 1 mL∙h^−1^ and the ES process was conducted for 2 h at room temperature. After the ES process, aluminum foils with keratin/gelatin bioplastic were stored for 4 days at room temperature for the drying process, which was necessary for completing the crosslinking.

The mechanical characterization of keratin-based bioplastic was carried out performing a uniaxial tensile test using a universal machine Zwick-Roell Z005 ProLine equipped with a 100 N load cell. Rectangular shaped specimens (length-to-width ratio 4:1) obtained from each electrospun sample were tested. Appendix A shows the testing setup. Samples were tested in triplicate, setting a strain rate of 10%/min of the initial length until failure. Load F [N] and elongation Δx [m] were recorded, and from the stress–strain curve the elastic modulus E [MPa], the failure stress σ_max_ [Pa], the corresponding failure strain ε_max_ [%], and toughness U [J·m^−3^] were calculated. Appendix A shows a typical stress-strain curve obtained from the uniaxial tensile test, from which the failure stress, strain, elastic modulus, and toughness were calculated.

The water permeability was evaluated using Darcy’s law with a custom-made setup already validated in another work [55], which allowed us to impose a constant pressure across the tested electrospun sample. Details on the setup are provided in Appendix A, while Appendix A shows a representative water permeability curve from which the volumetric flow rate and the Darcy permeability were determined.

The microscopic structure of dialyzed keratin and keratin-based electrospun was analyzed by FEI Quanta 450 ESEM FEG (Bruker), prior to metalation with platinum using a Leica EM ACE600 metallizer. The keratin-based electrospun materials were carefully evaluated for their fiber diameter and distribution. A quantitative diameter analysis was carried out by randomly measuring fibers in acquired images (at 60,000× magnification) using ImageJ software.

The TGA was performed on keratin-based electrospun materials with a TA Instruments Thermobalance model Q5000IR. Thermogravimetric measurements were taken at a rate of 10 °C∙min^−1^ and from 25 to 900 °C under nitrogen flow (25 mL∙min^−1^). The amount of sample in each measurement varied between 2 and 5 mg.

The DSC experiments on keratin-based electrospun materials were carried out by TA Instruments Discovery DSC model 250 under the nitrogen gas flow 50 mL∙min^−1^. The DSC was calibrated with Indium. For each electrospun material, 5–6 mg were weighted and were hermetically sealed into aluminum DSC pans. The samples were scanned by heating-cooling-heating cycles at a heating rate of 10 °C∙min^−1^ from 30 °C to 140 °C and cooling rate of 30 °C∙min^−1^ from 140 °C to 30 °C. An empty pan was used as a reference. The heat flow curve obtained in the first heating shows the glass transition process overlapped with the enthalpy relaxation peak (due to the self-assembly in structured domains), therefore, we evaluated the glass transition temperature (midpoint type: half height) from the second heating scan.

Infrared spectra were recorded by using a Perkin-Elmer Frontiers FTIR Spectrophotometer, equipped with a universal attenuated total reflectance (ATR) accessory and a triglycine sulphate TGS detector. Measurements on electrospun bioplastics, gelatin powder, and SK_IP_ were performed in ATR mode after background acquisition. For each sample, 128 scans were recorded, averaged, and Fourier-transformed to produce a spectrum with a nominal resolution of 4 cm^−1^.

A Principal Component Analysis (PCA) was performed by XLStat 2014.5.03 (Addinsoft, Paris, France) using the Pearson correlation coefficient matrix of the data. The variables taken into consideration were failure stress (σ_max_), failure strain (ε_max_), elastic modulus (E), toughness (T), permeability (K Darcy), fibers diameter, glass transition temperature (Tg), start degradation temperature (T_onset_), the ratio of optical densities of amide I and amide II bands (AII/AI), and of two bands characteristic of CH_2_ bending (1447/1409), and the concentrations of all fractions identified in the REK extracts (amino acids, SK_<1200_, SK_1200–7000_, SK_>7000_). Direct comparisons between different groups were performed using the Student’s *t*-test, and ANOVA, with a *p*-value < 0.05.

## 5. Conclusions

We have proposed an efficient, non-expensive, and green process to recover keratin from industrial wastes, which can be integrated in a one-pot process wherein raw keratin extract is directly used in ES to produce bioplastics, which are attractive for technological applications (e.g., bio packaging, purification/filtration membranes) in the framework of our circular economy and in waste recycling. The innovative one-pot process allows us to overcome the problem related to keratin aggregation that occurs during any purification process.

The results carried out on this study highlight the possibility of obtaining bioplastics with different performances by modulating the extraction parameters (time, MWA, CH), the soluble keratin, fully characterized in terms of fractions at different molecular weight (SK_<1200_, SK_1200–7000_, SK_>7000_ and amino acids), and the crosslinker (GPTMS) percent used.

In conclusion, with respect to previous studies on keratin extractions reporting total yields without any characterization of the material extracted in terms of proteins, peptides, and amino acids contents, [14,15,19] the novelties of this study include: (i) the optimization of the one-pot MAE process; (ii) the accurate characterization of all fractions of the REK solution obtained from the extraction process, with the yields referred to each fraction; (iii) the optimization of the ES process; and (iv) the characterization of the obtained bioplastic. Globally, this study has led to the delineation of a promising protocol for the industrial use and valorization of feather waste material.

## Data Availability

The raw data will be made available upon request.

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
