# Peer review of "One-Pot Process: Microwave-Assisted Keratin Extraction and Direct Electrospinning to Obtain Keratin-Based Bioplastic"

_ijms, 2021, doi:10.3390/ijms22179597_

Round 1

Reviewer 1 Report

This is a solid work about microwave-assisted keratin extraction and direct electrospinning to obtain keratin-based bioplastic. The idea is interesting and I think it will have some practical applications in the future. However, I still have some minor concerns before I can recommend it for publication, especially in scientific presentation of the data.

(1) The scale bars in SEM images are too small to read, such as Figure 1(a)-1(b) and Figure 2(b)-2(c). It's recommended to add a clear scale bar to the images.

(2) In Figure 3(a) and 3(b), the text font for the label is too small to read.

(3) In the introduction section, the authors give a brief introduction of electrospun materials from bio resources. However, I would like to suggest the authors add some discussions on other kinds of bio resource derived electrospun materials, such as lyocell fiber (DOI: 10.1016/j.jobab.2020.03.002) and cellulose (DOI:10.13360/j.issn.2096-1359.201911015) so that it could be more appealing to broad interest of readership.

Based on the above concerns, I suggest a minor revision.

Author Response

We thank the reviewers for the interesting suggestions.

As requested by both the reviewers, the English language and style were revised, simplifying the sentences, and removing redundant information, also with the help of professional software (Grammarly).

1) The scale bars in SEM images are too small to read, such as Figure 1(a)-1(b) and Figure 2(b)-2(c). It's recommended to add a clear scale bar to the images.

(2) In Figure 3(a) and 3(b), the text font for the label is too small to read.

As suggested by reviewer 1, the scale bars in Fig 1a-b, Fig 2b-c and the font label in Fig 3-b have been enlarged to improve comprehension.

(3) In the introduction section, the authors give a brief introduction of electrospun materials from bio resources. However, I would like to suggest the authors add some discussions on other kinds of bio resource derived electrospun materials, such as lyocell fiber (DOI: 10.1016/j.jobab.2020.03.002) and cellulose (DOI:10.13360/j.issn.2096-1359.201911015) so that it could be more appealing to broad interest of readership.

As suggested, we added in the manuscript a new paragraph on electrospun materials from bioresources. 

We cannot include the papers suggested by the reviewer in the bibliography because the first suggested is not specifically focused on electrospinning; the second paper is in Chinese, and fruitful analysis of this work is very difficult for us. We identified other works on biobased materials for supporting the discussion. 

Reviewer 2 Report

Review for ijms-1298942-v1

The work contains some useful information. However, one key concern is that too much descriptive analysis. Why selected gelatin? what was its contribution? What’s the contribution of microwave-assisted? For instance, Journal of Chemistry, 2019, 1326063.

The authors need to provide in-depth discussion to make this report more ‘scientific’. The authors emphasized that the key materials were gelatin. However, the authors did not provide key evidence that gelatin contributed special structure in resulting the properties. More discussion linked to the special structure/properties of gelatin is needed. For instance, Food Hydrocolloids, 94, 459-467; Food Hydrocolloids, 90, 9-18; Journal of Food Engineering, 239, 92-103. 

Figure 3 shows thermal properties while other figures/tables revealed mechanical, rheological properties etc. However, the integration of results of different parameters was very poor. The integration among these parameters would benefit elucidating the fundamental contribution of the components and logics behind these changes. For instance, Food Hydrocolloids, 99, 105317; Food Hydrocolloids, 75, 164-173.

Introduction: the first paragraph was too long, needs to be divided into at least two paragraphs.

Introduction: last paragraph: need to state the research objective clearly.

Tables: significant difference analysis among the groups should be conducted and labelled. Otherwise, what’s the use of these results? Any conclusion?

Table 5: add key conclusion/explanation from each item.

Scheme 1: not much useful. Suggest to use a diagram to integrate different results to make the story stronger. Please read the previous comments.

References: many of the references selected did not contribute much in discussion. More references related to the explanation of the results should be applied.

Author Response

We thank the reviewers for the interesting suggestions.

As requested by both the reviewers, the English language and style were revised, simplifying the sentences, and removing redundant information, also with the help of professional software (Grammarly).

The work contains some useful information. However, one key concern is that too much descriptive analysis. Why selected gelatin? what was its contribution? What’s the contribution of microwave-assisted? For instance, Journal of Chemistry, 2019, 1326063.

Following the reviewer's suggestion, we better highlighted the reasons for the use of gelatin in the introduction. We addressed a fully biocompatible and biodegradable biomaterial, and this concept has been also added in the abstract.

The extraction of keratin was performed by using MW-assisted extraction (MAE) with coaxial dipole antenna due to the good results obtained on the extraction of essential oils from orange peel wastes and different herbs, as reported in the introduction (page 2). We compared MAE and conventional heating (CH) extraction to assess the advantages/disadvantages of our MAE approach, mostly in terms of soluble keratin yields and final electrospun gelatine/keratin material properties.

The interesting work proposed by the referee Journal of Chemistry, 2019, 1326063 reports on the effect of MW and US irradiation on morphology and texture in the recrystallization stage of keratin, as well as to study the changes in the secondary structure (alpha or beta) and tertiary structure (peptide bonds) at different times and irradiation power. Keratin was extracted under acidic conditions (pH 4-5, T 50°C) milder than ours (acidic acid 70% and 104.6 °C for hours, MW or CH), and MW or US were applied for minutes in the recrystallization stage only, after washing keratin in ethanol/water solution. Differently, we didn’t purify keratin from the reaction bath, but we used “as it is” in acidic solution.

The study of the possible effects of microwave extraction on keratin structure is in progress in our group, as reported in the Discussion section (paragraph 3). Nonetheless, we believe that under the extreme experimental conditions we adopted, the different heating approach does not affect the structure of soluble keratin structure.

Nonetheless, the work suggested by the referee Journal of Chemistry, 2019, 1326063 has been added in Table 5 and the results are commented in the text.

To better clarify the reasons for the choice of MAE and CH in this work we modified the Introduction and we added the sentence “To evaluate the advantages/disadvantages of our MW assisted extraction process (MAE) approach mostly in terms of soluble keratin yields and final electrospun gelatine/keratin material properties, we compared MAE and conventional heating (CH) extraction.”. The effect of the heating mode of keratin extraction on the final electrospun gelatine/keratin material properties is reported on pages 2, 5, 7, 10, 14.

The authors need to provide in-depth discussion to make this report more ‘scientific’. The authors emphasized that the key materials were gelatin. However, the authors did not provide key evidence that gelatin contributed special structure in resulting the properties. More discussion linked to the special structure/properties of gelatin is needed. For instance, Food Hydrocolloids, 94, 459-467; Food Hydrocolloids, 90, 9-18; Journal of Food Engineering, 239, 92-103. 

As better specified in the introduction (page 3) our main goal was to produce keratin-based bioplastics by ES.

Keratin-only solutions are known to be not commonly electrospinnable, due to keratin's high conductivity and unsuitable rheological properties. For this reason, suitable blends of waste-derived keratin and porcine gelatin have been used in the present work with suitable amounts of crosslinking agent (GPTMS). Thus, the properties of gelatin-only electrospuns have been taken as a reference to study the effect of added keratin.  We aimed to observe the final of the biomaterial in the solid state. More in detail, we observed the effect of keratin content and blend composition on the bioplastic properties, performing a systematic evaluation on the variation of mechanical and thermal properties depending on the extraction mode and time, and GPTMS content. The structural evaluation of keratin-gelatin interaction of our biomaterial from a microscopical point of view was out of the manuscript's purposes/beyond the aim of this work.

As suggested by the reviewer, there are several works aimed to characterize the rheological and gelling properties of gelatin solutions, the effect of additives employed as co-gelling agents, and to investigate other gelling agents, alternatives to gelatin. We added a comment on this topic in the Introduction section.

Figure 3 shows thermal properties while other figures/tables revealed mechanical, rheological properties etc. However, the integration of results of different parameters was very poor. The integration among these parameters would benefit elucidating the fundamental contribution of the components and logics behind these changes. For instance, Food Hydrocolloids, 99, 105317; Food Hydrocolloids, 75, 164-173.

We agree with the reviewer on the importance of the integration among the parameters. A microscopic interpretation of the data was beyond the aim of this paper, while we thought it useful in this work to observe how the different material properties were correlated. For this reason, we have performed the principal component analysis (PCA), which allowed us an overall view of all data and to correlate different biomaterial properties (paragraph 2.4). In the PCA biplot the first component (F1) allowed the separation as a function of keratin content, and the second component (F2) as a function of GPTMS content, highlighting the predominance of these parameters in affecting most of the material properties. We added to the text a comment (page 9, above Fig.4) to highlight the PCA function to integrate different parameters.

Introduction: the first paragraph was too long, needs to be divided into at least two paragraphs.

Introduction: last paragraph: need to state the research objective clearly.

We thank the reviewer for these relevant remarks. Following his/her comments, the introduction was revised: the first paragraph was divided, and the last paragraph was modified to clearly explain the manuscript's purpose. Besides, the work was revised in its entirety, simplifying the sentences, and removing redundant information.

Tables: significant difference analysis among the groups should be conducted and labelled. Otherwise, what’s the use of these results? Any conclusion?

Due to the high amount of data collected, principal component analysis (PCA) was conducted to highlight statistically significant differences in the materials prepared. Besides, to improve the manuscript following the reviewer suggestions, we added a statistical comparison on each data group performing the Student test and ANOVA (p<= 0.05), where relevant. This point was specified in the main text, and the Material and Method section (last paragraph).

Table 5: add key conclusion/explanation from each item.

As suggested by the reviewer, each extraction procedure was discussed more in detail and compared with the others (below Table 5).

Scheme 1: not much useful. Suggest to use a diagram to integrate different results to make the story stronger. Please read the previous comments.

Scheme 1 helps in the comprehension of the manuscript. The extraction process involves several steps, and each fraction (defined with an abbreviation) was characterized with suitable techniques. Following the reviewer's comment, a final scheme (Scheme 1) to make the story stronger and integrate all the results was added at the end of the Discussion section.

References: many of the references selected did not contribute much in discussion. More references related to the explanation of the results should be applied.

References have been revised according to the reviewer's comment.

In particular, we added the following references:

- Materials Science & Engineering C 92 (2018) 969–982, Chem. Rev. 2019, 119, 5298−5415 have been added in the introduction to describe electrospun materials from bioresources.

- Journal of Chemistry, 2019, 1326063 has been added in Table 5 and commented in the text.

- Food Hydrocolloids, 94, 459-467; Food Hydrocolloids, 90, 9-18; Journal of Food Engineering, 239, 92-103 (suggested by the reviewer) has been added in the discussion section, to introduce the studies made on gelatin solutions and the effect of additives employed as co-gelling agents.

- Food Hydrocolloids, 99, 105317; Food Hydrocolloids, 75, 164-173 (suggested by the reviewer) have been added to introduce the investigation, reported in the literature other gelling agents, an alternative to gelatin.

- RSC Adv. 2017, 7, 9854–9861, doi:10.1039/c6ra25009b has been added to comment mechanical results.

- He, M.; Zhang, B.; Dou, Y.; Yin, G.; Cui, Y.; Chen, X. Nanofibers. RSC Adv. 2017, 7, 9854–9861, doi:10.1039/c6ra25009b has been added to comment on the keratin influence on the material stiffness.

Round 2

Reviewer 2 Report

The authors have addressed the questions quite well. There are no further comments. The current version is acceptable for publication.